# Sesbania Gum-Supported Hydrophilic Electrospun Fibers Containing Nanosilver with Superior Antibacterial Activity

**DOI:** 10.3390/nano9040592

**Published:** 2019-04-10

**Authors:** Shi Lan, Yaning Lu, Chun Li, Shuang Zhao, Naren Liu, Xianliang Sheng

**Affiliations:** College of Science, Inner Mongolia Agricultural University, Hohhot 010018, China; imaushilan@126.com (S.L.); shuijing_alp@163.com (Y.L.); lc_imdx@126.com (C.L.); nmgzhangchao@163.com (S.Z.); liunaren@163.com (N.L.)

**Keywords:** sesbania gum, polyacrylonitrile, silver nanoparticles, hydrophilic, electrospinning, antibacterial activity

## Abstract

In this contribution, we report for the first time on a new strategy for developing sesbania gum-supported hydrophilic fibers containing nanosilver using electrospinning (SG-Ag/PAN electrospun fibers), which gives the fibers superior antibacterial activity. Employing a series of advanced technologies—scanning electron microscopy, transmission electron microscopy, Fourier transform infrared spectroscopy, UV–visible absorption spectroscopy, X-ray photoelectron spectroscopy, and contact angle testing—we characterized the as-synthesized SG-Ag/PAN electrospun fibers in terms of morphology, size, surface state, chemical composition, and hydrophilicity. By adjusting the synthesis conditions, in particular the feed ratio of sesbania gum (SG) and polyacrylonitrile (PAN) to Ag nanoparticles (NPs), we regulated the morphology and size of the as-electrospun fibers. The fibers’ antibacterial properties were examined using the colony-counting method with two model bacteria: *Escherichia coli* (a Gram-negative bacterium) and *Staphylococcus aureus* (a Gram-positive bacterium). Interestingly, compared to Ag/PAN and SG-PAN electrospun fibers, the final SG-Ag/PAN showed enhanced antibacterial activity towards both of the model bacteria due to the combination of antibacterial Ag NPs and hydrophilic SG, which enabled the fibers to have sufficient contact with the bacteria. We believe this strategy has great potential for applications in antibacterial-related fields.

## 1. Introduction

In recent years, microorganism contamination arising from numerous emerging and re-emerging pathogens has posed an increasing threat to human health and has created an unprecedented challenge in antibacterial research [1]. To eliminate the looming crisis induced by pathogens, antibacterial materials able to prevent and control microorganism contamination have attracted significant interest. Many scientists, engineers, and front-line workers have contributed to the development of antibacterial materials that effectively eliminate or neutralize pathogens, while various sorts of antibacterial materials have emerged, such as oxidative halogens [2,3,4,5], metals and oxides [6,7,8], quaternary ammonium salts [9], peptides [10,11], guanidines [12,13], and *N*-halamines [14,15,16,17,18,19,20], among others. Unfortunately, pathogenic bacteria have developed drug resistance to most of the available traditional antibacterials through de novo mutation or by acquiring resistance genes from other organisms. Accordingly, there is an urgent need to explore new antibacterials that are different from traditional ones to fight against bacterial resistance.

Rapid developments in modern nanoscience and nanotechnology have provided promising alternatives. In contrast to traditional antibacterials, nanomaterials are less prone to induce bacterial resistance, because they are able to combine multiple antibacterial mechanisms into a single nanoentity [21]. To date, various kinds of nanomaterials, such as silver nanoparticles (Ag NPs) [22], gold nanorods [23], zinc oxide nanostructures [24], MXenes [25], and graphene oxide nanosheets [26,27] have been explored. In this diverse array, Ag NPs have drawn great interest and been widely used for medical purposes due to their effective antibacterial activity against a wide range of pathogenic bacteria [28]. Comprehensive literature surveys have demonstrated that compared to other antibacterial nanomaterials, Ag NPs are the most efficient and popular candidates [29,30,31]. However, their use is to a large extent restricted by their inter-particle aggregation, which may cause less antibacterial efficacy and even loss of antibacterial ability. In response to this shortcoming, Ag NPs have been loaded onto polymeric or inorganic nanocarriers, which not only prevent Ag NPs from inter-particle aggregation but also serve as carriers for the controlled release of antibacterial Ag NPs [32].

Many effective strategies have been proposed to load Ag NPs onto nanocarriers, such as electrospinning [33], template synthesis [34], a solvothermal technique [35], layer-by-layer assembly [36], and pressurized gyration [37,38,39,40]. Among them, electrospinning is one of the most popular. Since the development of electrospinning in the mid-1990s, it has generated bench research on the synthesis of functional nanomaterials and their practical applications in large-scale industrial production [41]. It is widely acknowledged that electrospinning is an effective and convenient technique to form functional electrospun fibers with sizes ranging from nanometers to a few microns, so electrospinning is highly suitable for the fabrication of antibacterial fibers [42]. Especially, electrospinning has been widely employed in the synthesis of antibacterial fibers containing Ag NPs [43]. It has been demonstrated that electrospinning is not suitable for mass production and manufacturing, it has been however explored widely in bench research on functional fibers in academic laboratories [44,45]. More significantly, electrospinning not only can simplify the synthetic process but also can yield powerful antibacterial activity due to the high surface-to-volume ratio of electrospun fibers. The synthesis of electrospun fibers containing antibacterial Ag NPs usually uses a polymer as a supporting matrix, such as cellulose acetate (CA), polyvinyl-alcohol (PVA), or polyacrylonitrile (PAN), to stabilize Ag NPs tightly within the electrospun fiber system. Of these polymer matrixes, PAN is the most popular [46,47,48]. Having PAN in an electrospun Ag/PAN system significantly improves the stability and long-term effectiveness of Ag NPs in antibacterial applications. However, when electrospun Ag/PAN is utilized in an aqueous environment, the presence of PAN may reduce the efficiency because its hydrophobic nature prevents good contact between the Ag NPs and the bacteria. This problem needs to be solved.

Sesbania gum (SG, shown in Figure 1A), extracted from sesbania plants, is a natural polysaccharide [49]. As a type of galactomannan, SG is made of glycosidic bonds linked to mannose as the main structures, and α(1→6) glycosidic bonds linked to galactose on the side chains [50]. It had been demonstrated that the physicochemical properties and chemical construction of SG are similar to those of other galactomannans [50]. A systematic literature survey indicated that previous studies of SG were primarily focused on its modification for subsequent use in industrial fields, mainly ore dressing, pharmaceuticals, textiles, printing, cosmetics, and petroleum, as well as others [51]. Most modifications have involved chemical reaction(s) on the side chains of SG, which is inconvenient, tedious, and costly. Using SG as an additive in practical applications without any modification is therefore an attractive prospect for basic research. Interestingly, SG is soluble not only in hot water but in cold water because of its chemical structure [52]. Hence, utilizing SG as a hydrophilic additive is a worthy avenue for exploration. However, the introduction of hydrophilic SG into a hydrophobic system to regulate the hydrophilic–hydrophobic balance has not previously been reported.

In this contribution, we for the first time propose a strategy to construct a novel antibacterial system in which hydrophilic SG was introduced into an Ag/PAN system using an electrospinning technique. As illustrated in Figure 1B, Ag NPs were synthesized via a chemical reduction reaction with Ag^+^ ions as the initial materials, and SG-supported hydrophilic electrospun fibers containing nanosilver were fabricated using electrospinning (SG-Ag/PAN electrospun fibers). After systematic characterization of the SG-Ag/PAN, we determined that the final products had a fibrous morphology, with Ag NPs encapsulated in the SG/PAN matrix-supported electrospun fibers. Their size and chemical composition were regulated by tuning the feed ratio of SG and PAN to Ag NPs. Because hydrophilic SG was present, the SG-Ag/PAN electrospun fibers displayed better antibacterial activity than Ag/PAN electrospun fibers. We believe our proposed strategy has important implications for antibacterial fields.

## 2. Materials and Methods

### 2.1. Materials

Silver nitrate (AgNO_3_), ethanol (EtOH), and *N*,*N*-dimethylformamide (DMF) were purchased from Sinopharm Chemical Reagent Co., Ltd. (Shanghai, China). Polyacrylonitrile (PAN) was obtained from the Tianjin Chemical Reagent Plant (Tianjin, China). Sesbania gum (SG) was available commercially from the Hongtu Plant Adhesive Factory (Qingdao, China). Graphene oxide (GO) was obtained from Nanjing XFNANO Materials Tech Co., Ltd. (Nanjing, China) Deionized water supplied by a Millipore system (Millipore Inc., Massachusetts, America) was utilized for all of the experiments. All the chemicals were used without purification.

### 2.2. Synthesis of Ag NP Dispersion

Synthesis of Ag NPs was carried out according to our previous report [53]. In a typical procedure, polyvinylpyrrolidone (PVP, 0.2 g) and AgNO_3_ (0.5 g) were added dropwise into 50 mL of *N*,*N*-dimethylformamide (DMF) under vigorous stirring. After 10 min of stirring, an Ag NPs dispersion was obtained. The as-synthesized Ag NPs were purified by several cycles of centrifugation and redispersed in water to remove residual PVP. The final Ag NP dispersion was obtained by dispersing Ag NPs in 50 mL of DMF.

### 2.3. Synthesis of SG-Ag/PAN Electrospun Fibers

The synthesis of SG-Ag/PAN electrospun fibers was accomplished using an electrospinning method. Typically, about 0.13 g of SG and 0.13 g of PAN were added into 50 mL of the Ag NP dispersion (see Section 2.3) and stirred overnight to obtain a transparent and achromatous precursor solution. A burette with a copper rod inserted to connect with a 15.0 kV source was filled with the precursor solution. Next, aluminum foil was fixed at a distance of about 15 cm from the burette tip, and electrospinning was performed at room temperature for about 30 min. After drying under a vacuum at 40 °C for 24 h, the SG-Ag/PAN electrospun fibers were collected onto aluminum foil. The feed ratio of SG and PAN to Ag NPs was successively changed from 1:1:4 to 3:1:4, 5:1:4, 7:1:4, and 9:1:4. For comparison, Ag/PAN and SG-PAN electrospun fibers were also prepared by the same approach.

### 2.4. Characterization

Scanning electron microscope (SEM) images were taken on a Shimadzu SSX-550 field emission scanning electron microscope (Shimadzu, Co. Ltd., Kyoto, Japan) at 15.0 kV. The as-electrospun samples were dispersed in ethanol with the assistance of an ice-water ultrasonic bath at 35 kHz, and a drop of the well-dispersed sample was cast repeatedly onto a piece of silicon wafer and air-dried. A thin gold coating was utilized to avoid charging during scanning, and a thorough microscopic study was carried out. The detailed morphology and size were examined with a Hitachi H-8100 transmission electron microscope (TEM, Hitachi, Ltd., Tokyo, Japan). As for SEM measurement, the as-electrospun samples for TEM were dispersed in ethanol with the assistance of an ice-water ultrasonic bath at 35 kHz. A drop of the well-dispersed sample above was cast repeatedly onto a specimen holder and air-dried. Fourier transform infrared (FTIR) spectra were recorded on a Thermo Nicolet Avatar 370 FTIR spectrometer. UV–visible absorption spectra (UV–vis, Hitachi, Ltd., Tokyo, Japan) were collected with a Hitachi U-3900H spectrophotometer in the 300–700 nm wavelength range. X-ray photoelectron spectra (XPS) were recorded with Mg Kα radiation.

### 2.5. Antibacterial Testing

Antibacterial testing was performed using the colony-counting method [54], with *Escherichia coli* (*E. coli*, ATCC 8099, Gram-negative bacterium) and *Staphylococcus aureus* (*S. aureus*, ATCC 6538, Gram-positive bacterium) as the two model bacteria. Typically, *E. coli* and *S. aureus* were grown overnight at 37 °C in a Luria-Bertani (LB) medium, then the bacterial cells were harvested by centrifugation, washed with phosphate-buffered saline (PBS), and diluted to a concentration of 1 × 10^6^ CFU/mL. Next, 50 μL of bacterial suspension was mixed with 0.45 mL of sample suspension (1 mg·mL^−1^) and incubated under constant shaking. After a certain period of contact time, the mixture was serially diluted, and 100 μL of each dilution was dispersed onto the LB growth medium. Survival colonies on LB plates were counted after incubation for 24 h at 37 °C. The colony-counting tests were carried out in triplicate. Bacterial survival was calculated according to the following Equation:
Bacterial Survival = (*B*/*A*) × 100%

where *A* is the number of colonies in the control and *B* is the number of bacterial colonies surviving after treatment with the sample.

### 2.6. Contact Angle Test

The contact angles were measured at 20 °C with the assistance of a static contact angle meter (KSV Instruments Ltd., Helsinki, Finland). Drops of water were deposited on the surface of a membrane sample using a manual dosing system that included a handheld 1 mL syringe. Side-view images of the drops were photographed at a rate of 10 frames per second. The contact angles were calculated automatically by fitting the drop shapes to the shape calculated from the Young–Laplace equation. The contact angles of each sample were measured in triplicate.

## 3. Results and Discussion

Figure 1B illustrates the fabrication procedure for the SG-Ag/PAN electrospun fibers. First, Ag NPs were easily synthesized by a chemical reduction reaction using Ag^+^ ions as the starting material. Ag NPs have potent antibacterial capability against a broad spectrum of pathogenic microorganisms, which is why we selected them for use in a complex system. Next, the as-synthesized Ag NPs, coupled with both SG and PAN, were suspended uniformly in the DMF system, and the mixture was transformed into fibers using electrospinning. Because of its wide availability and ease of use, PAN is the most popular and common support material involved in electrospinning-based advanced synthesis. However, the antibacterial efficacy of PAN-based electrospun fibers is to some extent restricted by PAN’s hydrophobicity. So hydrophilic SG was selected as the third component in the complex system, to enhance the hydrophilicity of the Ag/PAN fibers and thereby increase the contact between Ag NPs and bacteria. In this way, an optimal combination of these three components yielded a product with superior antibacterial activity (Figure 1C).

The morphological characteristics of the as-prepared products were examined using SEM. As shown in Figure 2A, SG-Ag/PAN displays an obvious fiber-like morphology with randomly oriented, straight, and continuous features. Most fibers within the selected range show good growth in one dimension (1D), displaying a quite long and uniform 1D appearance. The surface states of the SG-Ag/PAN fibers were then investigated using a magnified SEM image (as shown in Figure 2B). The surface of each fiber is fairly smooth, and there is no fiber–fiber aggregation. From the two SEM images, it is difficult to determine whether SG and Ag NPs were located inside or on the surface of the PAN fibers. As the insert image in Figure 2A (red curve) shows, the SG-Ag/PAN fibers had a narrow size distribution of 589.3 ± 143 nm. In addition, the TEM images in Figure 2C reveal a quite uniform distribution, with no significant cracks or irregularities. More detailed information is captured from the magnified TEM image in Figure 2D. Many small black dots (red arrows) are visible and well distributed among the fiber supports (gray background), suggesting that nanosized silver particles with a dot-like appearance were widely scattered throughout the PAN and SG mixture. Significantly, there is no visible particle–particle aggregation of Ag NPs, suggesting that the polymeric matrix SG-PAN prevented Ag NPs from aggregating. As a result, SG-Ag/PAN has the appearance of fibers with interiors decorated by dots. We accordingly conclude that electrospinning resulted in SG-Ag/PAN fibers with good Ag NP dispersion within the SG-modified PAN system.

The presence of Ag NPs in the as-synthesized fibers was further evidenced by UV–visible absorption spectra (UV–vis). As shown in Figure 3, an absorption peak maximum of 412 nm is observable in the UV–vis curve of pristine Ag NPs, which is highly consistent with previous reports [55]. Similarly, the SG-Ag/PAN electrospun fibers have one prominent peak at around 411 nm, assignable to the Ag NPs component, suggesting that they were thoroughly encapsulated within the complex fibers. More interestingly, when the UV–vis spectra of Ag/PAN electrospun fibers (not given) is compared with that of the SG-Ag/PAN fibers, no difference is detectable, suggesting that the introduction of SG into the Ag/PAN system had no impact on the Ag NPs. Also, the SG-Ag/PAN fibers show no other absorption within the 300 to 700 nm range, suggesting that the as-electrospun products were relatively pure.

Detailed information about the chemical compositions of the SG-Ag/PAN electrospun fibers was gathered with the assistance of XPS. As can be seen from the resulting spectra (Figure 4A–D), the SG-Ag/PAN fibers exhibited three main peaks for C 1s, N 1s, and O 1s, centered at 285 eV, 400 eV, and 532 eV, respectively [56]. As elemental markers for PAN and SG, the co-existence of N 1s and O 1s proves that the combined polymer support was composed of SG and PAN. Interestingly, it is difficult to discern the specific signal of Ag 3d, indicating that the nanosilver was dispersed deep inside the fibers rather than near the surface. This is further evidence of the successful encapsulation of nanosilver in the SG–PAN mixed system.

For most antibacterial nanomaterials, especially contact-dependent ones, the material’s size has an important impact on its antibacterial activity: the smaller the material, the stronger its bactericidal activity [57]. After confirming the validity of the electrospinning-based strategy to fabricate SG-Ag/PAN electrospun fibers, we examined tuning of the fibers’ size by changing the feed ratio of PAN and SG to Ag NPs. In our step-by-step feed ratio-controlled synthesis, the feed ratio of PAN and SG to Ag NPs was varied from 1:1:4 to 3:1:4, 5:1:4, 7:1:4, and 9:1:4, while other parameters were kept constant. Figure 5 and Figure 6 display the SEM images and corresponding size distributions of the as-electrospun fibers prepared with different feed ratios. Interestingly, the five fibers had similar morphologies (fiber-like appearances) but different sizes. The size increased from 285.2 ± 67 nm to 321.2 ± 82 nm, 393.7 ± 101 nm, 537.7 ± 123 nm, and 589.3 ± 143 nm when the feed ratio increased from 1:1:4 to 3:1:4, 5:1:4, 7:1:4, and 9:1:4, suggesting that increasing the feed ratio increased the size of the SG-Ag/PAN fibers. Consequently, we conclude that the fiber size can be modulated simply by controlling the feed ratio of PAN and SG to Ag NPs, although it should be noted that the size distribution broadened as the feed ratio increased from 1:1:4 to 9:1:4. Notably, all five products had a fiber-like shape and a smooth surface without any agglomeration or breakage, suggesting that changing the feed ratio had no impact on their morphology.

We also investigated the influence of feed ratio on the composition and content of SG-Ag/PAN electrospun fibers. Five fibers were synthesized with feed ratios of 1:1:4, 3:1:4, 5:1:4, 7:1:4, or 9:1:4, and the FTIR spectra of the corresponding products were analyzed. As shown in Figure 7, all five products exhibit two characteristic peaks at 1034 cm^−1^ and 2243 cm^−1^, attributable to C–O bending and C≡N stretching, respectively [58,59]. The coexistence of these peaks is a good indicator that SG was well combined with PAN. When the intensities of the characteristic peaks were compared, some difference was evident. Generally, the signals attributable to the C≡N peak (indicating PAN) gradually increased as the feed ratio rose from 1:1:4 to 9:1:4, suggesting increasing PAN content in the SG-Ag/PAN fibers. In contrast, the intensity of the C–O peak weakened as the feed ratio rose from 1:1:4 to 9:1:4, confirming a decrease in SG content. Hence, we can conclude that changing the feed ratio altered the composition of the final product.

Aside from examining the effect of feed ratio on fiber size and composition, we investigated the antibacterial activity of Ag/PAN, SG-Ag/PAN, and SG-PAN, using *E. coli* and *S. aureus* as two models in a bacterial concentration of 10^5^ CFU·mL^−1^ [60,61,62,63,64,65,66,67]. In our antibacterial tests, a well-known antibacterial nanomaterial, graphene oxide (GO) [68,69,70], was also examined as a comparative control. In Figure 8A, surviving bacterial colonies appear as small white dots on the yellow culture plates. On the control plates, both *E. coli* and *S. aureus* show dense survival colonies, illustrating the robust growth of the two selected strains in the absence of the electrospun fiber samples. After the bacterial suspensions had had contact with the samples for 120 min and then were incubated for 12 h, dense bacterial survival was detected on the culture plates corresponding to SG-PAN, illustrating that both SG and PAN had almost no antibacterial activity against the two selected strains. In contrast, no survival colonies were found in the presence of GO, Ag/PAN, or SG-Ag/PAN, suggesting their excellent bactericidal abilities against both *E. coli* and *S. aureus* under the experimental conditions. When the Ag/PAN, SG-Ag/PAN, and SG-PAN samples are compared, it is clear that the Ag NPs were the actual antibacterial component, rather than SG or PAN.

Although the success of both Ag/PAN and SG-Ag/PAN against two bacterial strains was confirmed by contrasting them with the control, the difference between Ag/PAN and SG-Ag/PAN electrospun fibers could scarcely be detected from the photographs of the culture plates. Subsequently, we carried out antibacterial kinetic tests using *E. coli* and *S. aureus*. Figure 8B,C presents the corresponding results in terms of calculated survival. *E. coli* and *S. aureus* maintained a constant survival level close to 100% in the presence of SG-PAN, demonstrating that neither SG nor PAN had bactericidal capability against the selected strains throughout the entire aging period (from 0 to 120 min). In comparison to SG-PAN, the Ag/PAN electrospun fibers demonstrated antibacterial activity, resulting in only 7% survival for *E. coli* and 0.3% survival for *S. aureus* after 120 min. More significantly, when the two strains were exposed to the SG-Ag/PAN fibers for only 60 min, almost no survival was evident for both *E. coli* and *S. aureus*. Comparison between Ag/PAN and SG-Ag/PAN demonstrated that the SG-Ag/PAN fibers had greater bactericidal activity than the Ag/PAN fibers, confirming the contribution of SG to the antibacterial action of SG-Ag/PAN. The results of the kinetic test further demonstrated that the active component in the SG-Ag/PAN fibers was the Ag NPs rather than PAN or SG, although the addition of SG into the Ag/PAN system enhanced its antibacterial efficacy.

We speculate that the most plausible explanation for the SG-induced antibacterial reinforcement is the hydrophilicity of SG, which ensured the good dispersion of Ag/PAN fibers in the bacterial suspension and consequently allowed the Ag NPs to come into full contact with the bacteria, yielding excellent antibacterial activity. To test this hypothesis, we examined the hydrophilic and hydrophobic properties of SG-Ag/PAN electrospun fibers prepared with different feed ratios: 1:1:4, 3:1:4, 5:1:4, 7:1:4, and 9:1:4. The fibers’ hydrophilic and hydrophobic properties were gauged using the contact angle test. As shown in Figure 9, the contact angles of the five SG-Ag/PAN fibers synthesized at feed ratios of 1:1:4, 3:1:4, 5:1:4, 7:1:4, and 9:1:4 were 15.5°, 22.1°, 29.0°, 30.1°, and 37.8°, respectively, all of which are lower than the 52° contact angle of Ag/PAN fibers; this indicates that the SG-Ag/PAN fibers were more hydrophilic than the Ag/PAN fibers [71,72]. Comparing the hydrophilicity of the Ag/PAN and SG-Ag/PAN fibers suggests that the addition of SG made the electrospun fibers more hydrophilic. A detailed comparison of the five SG-Ag/PAN fibers synthesized with different feed ratios showed that the contact angle rose as the PAN content increased, demonstrating that the lower the PAN content in the SG-Ag/PAN, the more hydrophobic they were. Accordingly, introducing SG into the Ag/PAN system made the SG-Ag/PAN fibers hydrophilic, which facilitated their contact with bacteria in water. As a result, the SG-Ag/PAN fibers were more effective antibacterial agents than the Ag/PAN fibers.

## 4. Conclusions

In summary, novel three-component antibacterial nanomaterials in the form of SG-Ag/PAN fibers were successfully synthesized using electrospinning. The as-synthesized SG-Ag/PAN electrospun fibers were systematically characterized using SEM, TEM, XPS, UV–visible spectra, and FTIR spectra. After confirming that the final products had a fiber-like morphology and a smooth surface, we regulated their size and chemical composition by tuning the feed ratio of PAN and SG to the Ag NPs. The antibacterial activity of the SG-Ag/PAN was evaluated using the colony-counting method with *E. coli* and *S. aureus* as two representative bacterial models, and the corresponding results demonstrated that owing to the presence of hydrophilic SG, the SG-Ag/PAN fibers had better antibacterial capability than the Ag/PAN fibers. Our study therefore reveals that SG enhances Ag/PAN’s antibacterial activity.

## Figures and Tables

**Figure 1 nanomaterials-09-00592-f001:**
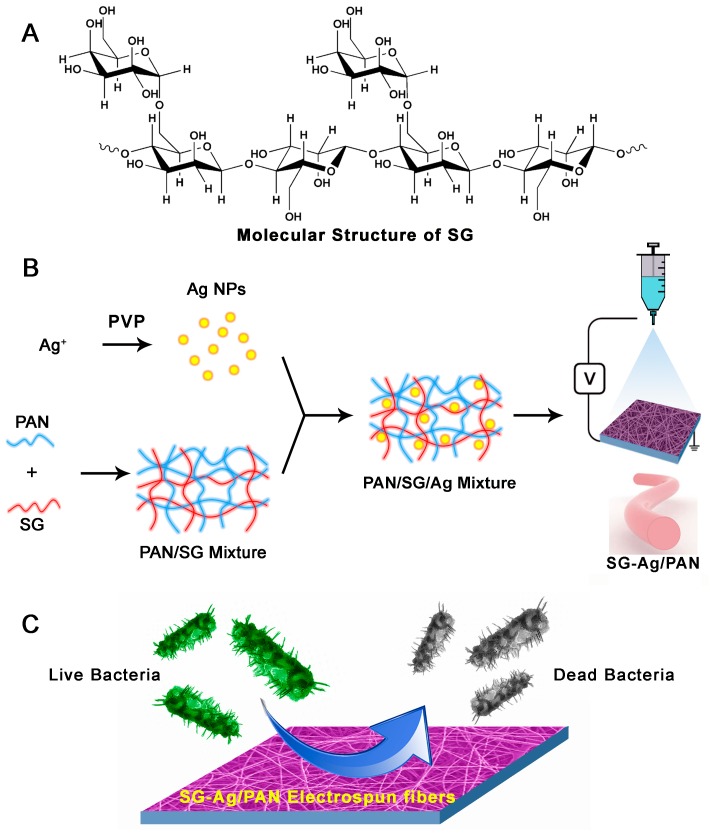
(**A**) Molecular structure of sesbania gum (SG). (**B**) Schematic illustration of SG-Ag/PAN electrospun fiber synthesis using electrospinning. (**C**) Schematic depiction of the antibacterial action of SG-Ag/PAN electrospun fibers against bacteria.

**Figure 2 nanomaterials-09-00592-f002:**
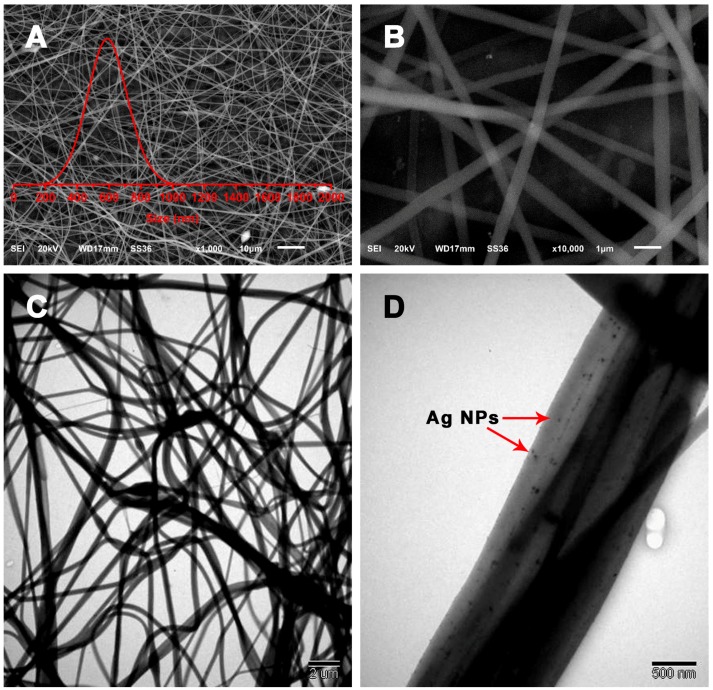
(**A**,**B**) SEM images, (insert) size distribution, and (**C**,**D**) TEM images of SG-Ag/PAN electrospun fibers at different magnifications.

**Figure 3 nanomaterials-09-00592-f003:**
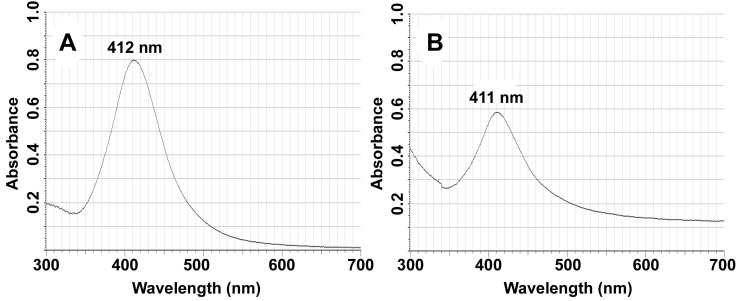
UV–visible absorption spectra of (**A**) Ag NPs and (**B**) SG-Ag/PAN electrospun fibers.

**Figure 4 nanomaterials-09-00592-f004:**
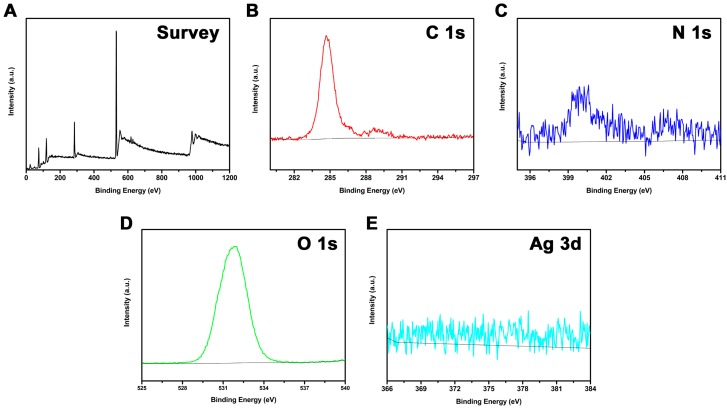
(**A**) XPS survey spectrum, (**B**) C 1s spectrum, (**C**) N 1s spectrum, (**D**) O 1s spectrum, and (**E**) Ag 3d spectrum of SG-Ag/PAN electrospun fibers.

**Figure 5 nanomaterials-09-00592-f005:**
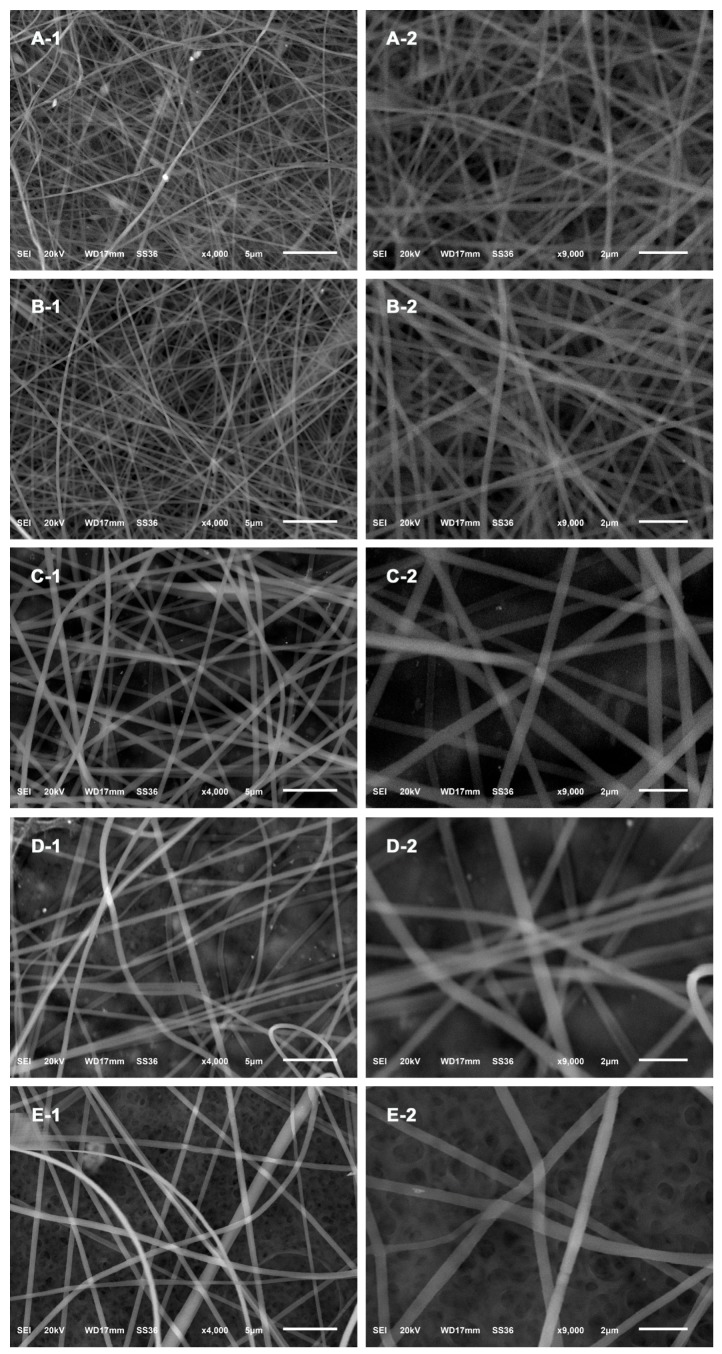
SEM images of SG-Ag/PAN electrospun fibers prepared with different feed ratios of PAN and SG to Ag NPs: (**A**) 1:1:4, (**B**) 3:1:4, (**C**) 5:1:4, (**D**) 7:1:4, and (**E**) 9:1:4.

**Figure 6 nanomaterials-09-00592-f006:**
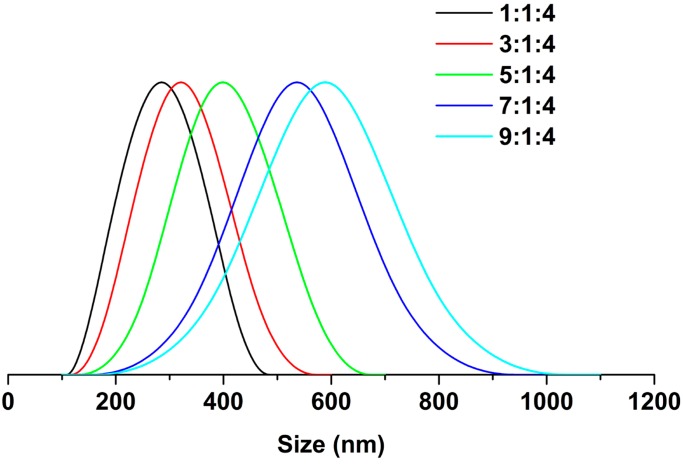
Size distribution of SG-Ag/PAN electrospun fibers prepared with different feed ratios of PAN and SG to Ag NPs: 1:1:4, 3:1:4, 5:1:4, 7:1:4, and 9:1:4.

**Figure 7 nanomaterials-09-00592-f007:**
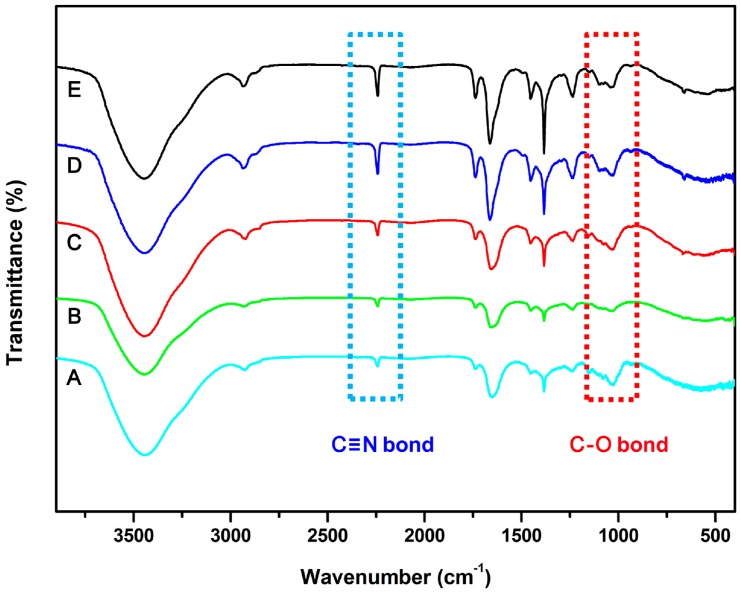
FTIR spectra of SG-Ag/PAN electrospun fibers prepared with different feed ratios of PAN and SG to Ag NPs: (**A**) 1:1:4, (**B**) 3:1:4, (**C**) 5:1:4, (**D**) 7:1:4, and (**E**) 9:1:4.

**Figure 8 nanomaterials-09-00592-f008:**
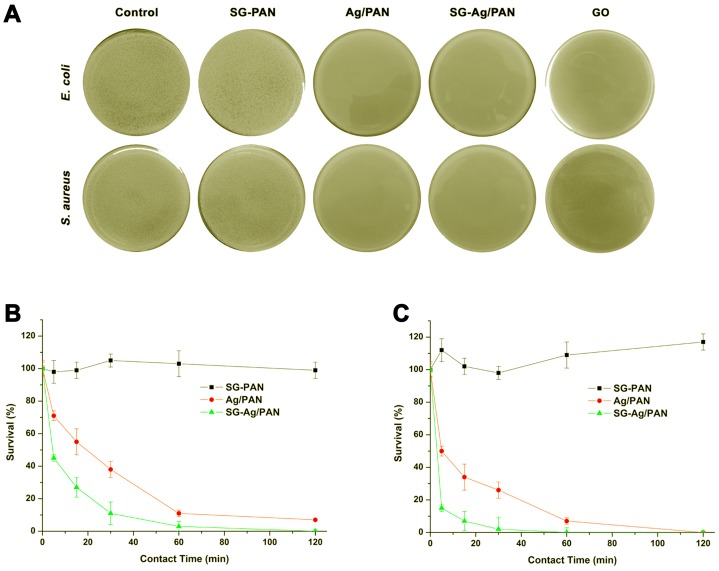
(**A**) Photographs of the bacterial culture plates of *E. coli* and *S. aureus* after 120 min of exposure to nothing (the control) or to SG-PAN, Ag/PAN, or SG-Ag/PAN electrospun fibers, or to graphene oxide (GO). Antibacterial kinetic tests for SG-PAN, Ag/PAN, and SG-Ag/PAN electrospun fibers against (**B**) *E. coli* and (**C**) *S. aureus*.

**Figure 9 nanomaterials-09-00592-f009:**
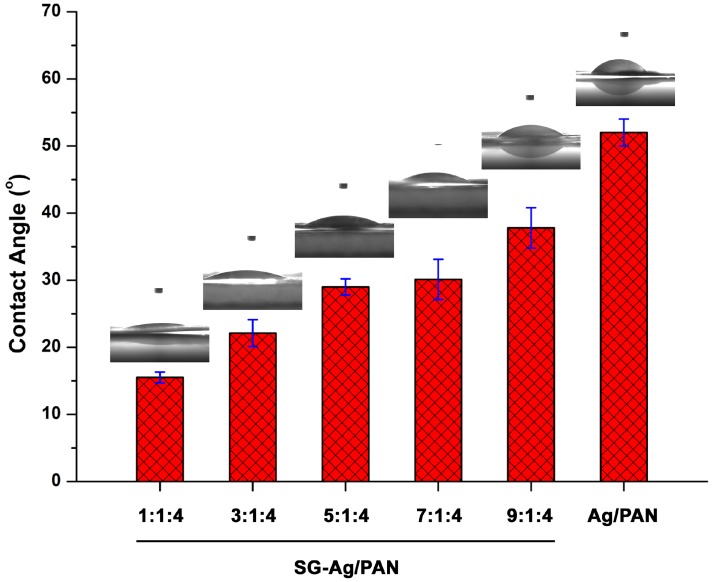
Contact angles of Ag/PAN and SG-Ag/PAN electrospun fibers prepared with different feed ratios of PAN and SG to Ag NPs.

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
