# Peer review of "Sesbania Gum-Supported Hydrophilic Electrospun Fibers Containing Nanosilver with Superior Antibacterial Activity"

_nanomaterials, 2019, doi:10.3390/nano9040592_

Reviewer 1 Report

I was on the border of rejecting this manuscript as it has major problems, however I have given the authors' another chance. Major problems:

Lacks full details of electrospinning

The evidence that the particles/fibers are nano are non-existant. SEM figures and also size distributions in Fig 5 need to be of higher resolution and clearer. Need mean and std dev. What is nano to the authors?

Why were they made this way? - see recent review by Heseltine et al. in Macromol Mater Eng, Sept 2018. Compare and contrast.

The antibac results are unconvincing, no comparisons are made - see recent literature by Ciric et al., Matharu et al. , Zewen Xu et al. and compare.

Many minor flaws....., but we can cross these later. English needs a thorough polish throughout.

Author Response

Reviewer #1:

I was on the border of rejecting this manuscript as it has major problem, however I have given the authors' another chance. Major problems:

Comment 1: Lacks full details of electrospinning.

Response : According to the reviewer suggestion, more detailed information regarding to the electrospinning have been added in the revised manuscript.

Comment 2: The evidences that the particles/fibers are nano are non-existant. SEM figures and also size distributions in Fig 5 need to be of higher resolution and clearer. Need mean and std dev. What is nano to the authors?

Response : Thanks for your suggestion. As the reviewer said, the as-synthesized products, SG-Ag/PAN electrospun fibers, are not nano in size. It is defined that nano size is always lower than 100 nm, and sizes between 100 nm and 1000 nm belong to submicron. The corresponding descriptions in whole text have been revised. Most significantly, the title has also been revised as ‘Sesbania Gum-Supported Hydrophilic Electrospun Fibers Containing Nanosilver with Superior Antibacterial Activity’ in our revised manuscript.

Besides, a clear and revised Figure 5 with high resolution has been provided in the revised manuscript. Also the mean sizes and standard deviations have also been added in the corresponding discussion parts.

Comment 3: Why were they made this way? - see recent review by Heseltine et al. in Macromol Mater Eng, Sept 2018. Compare and contrast.

Response : A systematic survey and comprehensive assessment have demonstrated that nanofibers could be facilely manufactured on a suitable scale using the pressurized gyration (PG) method. Clearly, PG, as a new solvent-based nanofabrication strategy, was first developed by Mahalingam and Edirisinghe, and has been proven to be an effective pathway to achieve functional fibers. Some typical literatures (Ref. 37-40) have been cited in our revised manuscript. Considering that the limited experimental conditions we have now, we however can hardly fabricate three component products SG-Ag/PAN using PG method. However, PG might be a better choice in preparing functional polymer fibers in our future studies.

Additionally, many advanced techniques, such as template synthesis, solvothermal method, electrospinning technique, and pressurized gyration method, have been explored to produce antibacterial nanostructures, with electrospinning technique being one popular candidate. After the term electrospinning was coined in mid 1990s, electrospinning technology has gained a great achievement because it is unique both in terms of ease of operation and cost-effectiveness. By the assistance of electrospinning technique, numerous antibacterial materials have been successfully electrospun. Electrospinning not only can simplify the synthetic process but also can yield high antibacterial activity due to the high surface-to-volume ratio of electrospun nanofibers. These are why we select electrospinning in our present study to fabricate SG-Ag/PAN fibers, and more importantly, our group has been committed to electrospinning-based strategy to design and construct multifunctional and smart fibers for several years.

Comment 4: The antibac results are unconvincing, no comparisons are made - see recent literature by Ciric et al., Matharu et al. , Zewen Xu et al. and compare.

Response : According to the reviewer suggestion, a typical antibacterial nanomaterial, graphene oxide (GO), has been selected as comparative control, whose antibacterial activity has been examined as well using the colony-counting method in our revised manuscript. Also, some representative literatures (Ref. 58-68) have been cited in the revised manuscript.

Comment 5: Many minor flaws....., but we can cross these later. English needs a thorough polish throughout.

Response : According to the reviewer suggestion, the whole text has been revised and checked carefully, and then English in whole text has been thoroughly polished by two native English speaking experts.

Reviewer 2 Report

This paper reported the preparation and characterization of silver nanoparticles containing PAN electrospun nanofibers based composite, where the silver nanoparticles were supported by sesbania gum polysaccharide. The authors conduct a series of measurements such as TEM, SEM, FTIR, XPS, UV-VIS, contact angle and antibacterial measurement. It is considered the information delivered from this paper would be attractive to researchers who are interested in creation of antibacterial surfaces and coating materials.

My opinion is that this article after major revisions can be published in Nanomaterials:

1.) The description of Ag NPs dispersion’s synthesis is imprecise! Why formed AgNPs during the synthesis, what was the reducing agent? The type of contact angle meter is also missing in this section.

2.) Did the author measure the size and size distribution of AgNPs after the synthesis? Did it change after the electrospinning process?

3.) The paper only discusses the role of SG and PAN in the system, however, beside these polymers, there was PVP, as well. Moreover, the amount of PVP was higher (0.2 g), than the amount of SG and PAN (0.13-0.13 g). Why did the authors omit the presence of this component?

4.) line 174: “As can be also seen in Figure 2A, the SG-Ag/PAN fibers had a narrow size distribution, with an average size of 589 nm.” Please complete the missing size distribution curve.

5.) What kind of interaction are between the polymer fibers and the AgNPs?

6.) Fig. 5: enlarge please the scale of the size distribution curve

7.) line 79: “SG is soluble in cold as well as hot water because of its chemical structure [46].” How his water solubility affect the long- term usability and antibacterial activity of the samples? Did the AgNPs remain firmly attached on the surface of fibers?

8.) What it the active (antibacterial) agent in this system? As you know, the surface immobilized AgNPs release silver ions into the environment (e.g.: Tallósy et. el; Environ Sci Pollut Res (2014) 21:11155–11167). Please discuss the antibacterial mechanism in this regard. 

Author Response

Reviewer #2:

This paper reported the preparation and characterization of silver nanoparticles containing PAN electrospun nanofibers based composite, where the silver nanoparticles were supported by sesbania gum polysaccharide. The authors conduct a series of measurements such as TEM, SEM, FTIR, XPS, UV-VIS, contact angle and antibacterial measurement. It is considered the information delivered from this paper would be attractive to researchers who are interested in creation of antibacterial surfaces and coating materials. My opinion is that this article after major revisions can be published in Nanomaterials:

Comment 1: The description of Ag NPs dispersion’s synthesis is imprecise! Why formed Ag NPs during the synthesis, what was the reducing agent ? The type of contact angle meter is also missing in this section.

Response : In our synthetic strategy, the Ag nanoparticles (Ag NPs) were easily synthesized by a chemical reduction reaction using polyvinylpyrrolidone (PVP) as both reducing agent and stabilizer. In addition, the type of contact angle employed in the present work is static contact angle, and the corresponding description has been amended in our revised manuscript.

Comment 2: Did the author measure the size and size distribution of Ag NPs after the synthesis ? Did it change after the electrospinning process?

Response : Actually, the size and size distribution of Ag NPs before and after electrospinning treatment have been compared, and the final results demonstrated that size and size distribution have almost no significant change after electrospinning treatment. Therefore, we can conclude that the electrospinning employed in this study has no impact on the size of the Ag NPs.

Comment 3: The paper only discusses the role of SG and PAN in the system, however, beside these polymers, there was PVP, as well. Moreover, the amount of PVP was higher (0.2 g), than the amount of SG and PAN (0.13-0.13 g). Why did the authors omit the presence of this component?

Response : As reviewer said, it is clear that the amount of PVP is higher than those of SG and PAN. However, PVP that selected in our synthetic strategy as reducing agent and stabilizer was employed in 2.2 Synthesis of Ag NPs Dispersion rather than in electrospinning process. After the Ag NPs dispersion was well prepared, the as-synthesized Ag NPs were purified by several cycles of centrifugation and redispersion in water. So the residual PVP has been eliminated, and only those PVP immobilized on surface of Ag NPs has been involved in 2.3 Synthesis of SG-Ag/PAN Electrospun Fibers. In comparison to PVP immobilized on surface of Ag NPs, PAN and SG contribute more to the antibacterial system, and therefore the role of SG and PAN in regulating the antibacterial properties has been discussed systematically in our study.

In addition, the experimental sections of 2.2 Synthesis of Ag NPs Dispersion and 2.3. Synthesis of SG-Ag/PAN Electrospun Fibers have been revised and presented in a clear manner in our revised manuscript.

Comment 4: Line 174: “As can be also seen in Figure 2A, the SG-Ag/PAN fibers had a narrow size distribution, with an average size of 589 nm.” Please complete the missing size distribution curve.

Response : According to the reviewer suggestion, the image showing size distribution of SG-Ag/PAN fibers (the insert in Figure 2A) has been added in the revised manuscript. Also the corresponding figure caption has been revised in our revised manuscript.

Comment 5: What kinds of interaction are between the polymer fibers and the Ag NPs?

Response : The Ag NPs have distributed well into the polymer matrix of PAN and SG by the assistance of some typical interactions between Ag NPs with polymers, such as coordination interaction, electrostatic interactions, etc., which make Ag NPs stable and long-term effective in bacteria-killing.

Comment 6: Fig. 5: enlarge please the scale of the size distribution curve.

Response : According to the reviewer suggestion, an enlarged version of size distribution (Figure 6) has been given in the revised manuscript.

Comment 7: Line 79: “SG is soluble in cold as well as hot water because of its chemical structure [46].” How his water solubility affect the long- term usability and antibacterial activity of the samples? Did the Ag NPs remain firmly attached on the surface of fibers?

Response : In our designed SG-Ag/PAN electrospun system, SG was mixed well with PAN to acting as polymer matrix, and the pre-synthesized Ag NPs could be added and stabilized into the SG-PAG bi-component system. Obviously, the pristine SG molecules could swell completely into water system due to its good water solubility. However, the PAN can entwine with SG in the SG-Ag/PAN system, and thus the presence of PAN to some extent prevents SG from infinite swelling, as a result the Ag NPs could be fixed tightly into the electrospun fibers to endow long-term antibacterial function.

Comment 8: What it the active (antibacterial) agent in this system? As you know, the surface immobilized Ag NPs release silver ions into the environment (e.g.: Tallósy et. el; Environ Sci Pollut Res (2014) 21:11155-11167). Please discuss the antibacterial mechanism in this regard.

Response : Recent studies demonstrated that silver nanoparticles (Ag NPs) draw interest for their unique and powerful antibacterial capabilities against a wide spectrum of pathogenic bacteria. As evidenced by our antibacterial tests (Figure 8), the antibacterial component of the as-synthesized SG-Ag/PAN electrospun fibers is Ag NPs rather than SG and PAN. To date, numerous studies have investigated the mechanisms for the antibacterial action of Ag NPs-based nanomaterials. Basically, their antibacterial mechanisms of can be classified into following types: (1) endowing reactive oxygen species-dependent oxidation stress; (2) causing irreversible damage to DNA; (3) binding -SH on the surfaces of proteins; (4) electrostatic interactions; and (5) combining two or more mechanisms given above. It was demonstrated that a single antibacterial mechanism mentioned above may not be potent enough for the efficient antibacterial action of SG-Ag/PAN electrospun fibers. It has been speculated that a combination of two or three mechanisms should be more responsible for antibacterial efficacy at the same time.

Based on those systematic bench investigations, it was acknowledged that the antibacterial materials constituting Ag NPs are more popular compared with other metallic nanoparticles. Nevertheless, practical applications of Ag NPs are often hampered by oxidization reactions, which may cause aggregation and even loss of antibacterial activity. Taking these shortcomings into consideration, polymer matrix (i.e., SG-PAN) employed herein can protect Ag NPs from aggregation, and because of the polymer matrix, Ag NPs could be fixed tightly into the three component system, as a result the final products SG-Ag/PAN can realize long-term antibacterial activity.

Reviewer 3 Report

The manuscript entitled "Sesbania gum-supported hydrophilic nanofibers containing nanosilver with superior antibacterial activity" describes a new strategy for developing fibers with superior antibacterial activity. The number of materials with antimicrobial activity has increased considerably over last few years. The continuous use of antimicrobial compounds can lead to bacterial resistance. To minimize these risks, there are currently a high demand for antimicrobial materials with non toxic natural compounds and environmentally friendly. For these reasons the results of this work will be of great interest for the scientific community. I recommend the publication of this article in Nanomaterials, pending some revisions.

- The authors should explain the preparation of the samples for SEM and TEM

- Figure 5 must be rearranged because the scale and magnification of the images and size distribution of nanofibers are not visible

- The authors should improve the images of figure 7A

Author Response

Reviewer #3:

The manuscript entitled "Sesbania gum-supported hydrophilic nanofibers containing nanosilver with superior antibacterial activity" describes a new strategy for developing fibers with superior antibacterial activity. The number of materials with antimicrobial activity has increased considerably over last few years. The continuous use of antimicrobial compounds can lead to bacterial resistance. To minimize these risks, there are currently a high demand for antimicrobial materials with non toxic natural compounds and environmentally friendly. For these reasons the results of this work will be of great interest for the scientific community. I recommend the publication of this article in Nanomaterials, pending some revisions.

Comment 1: The authors should explain the preparation of the samples for SEM and TEM.

Response : According to the reviewer suggestion, more detailed information about the preparation of samples for SEM and TEM measurements have been provided in the revised manuscript.

Comment 2: Figure 5 must be rearranged because the scale and magnification of the images and size distribution of nanofibers are not visible.

Response: Based on the reviewer suggestion, Figure 5 has been rearranged in the revised manuscript. In addition, the size distribution has been rearranged as Figure 6 in our revised manuscript.

Comment 3: The authors should improve the images of figure 7A.

Response : According to the reviewer’s suggestion, Figure 7A has been revised in the revised manuscript.

Round  2

Reviewer 1 Report

The authors have made a good attempt to respond/revise. However extensive English editing is needed, one cannot have phrases like "Thanks to its ease of operation, universality, and cost-effectiveness, electrospinning has been .....

Also electrospinning is too glorified, it must be clearly stated that it is not suitable for mass production/manufacturing:

See http://electrospintech.com/production-level.html#.XKBu8FVKjIU

Cite paper by Luo et al., in it.

Author Response

Reviewer #1:

Comment 1: The authors have made a good attempt to respond/revise. However extensive English editing is needed, one cannot have phrases like "Thanks to its ease of operation, universality, and cost-effectiveness, electrospinning has been

Response : Thanks for your kind help ! A further English editing has been done via a professional English service.

Comment 2:Also electrospinning is too glorified, it must be clearly stated that it is not suitable for mass production/manufacturing:

See http://electrospintech.com/production-level.html#.XKBu8FVKjIU

Cite paper by Luo et al., in it

Response :According the the reviewer suggestion, the description ‘It is demonstrated that electrospinning is not suitable for mass production and manufacturing, it is however explored widely in bench research on functional fibers in academic laboratories.’ has been added in the revised manuscript. In addition, two representative literatures (Ref. 44,45, authored by Luo et al.,) have been cited in our revised manuscript.

Reviewer 2 Report

The authors have satisfactorily responded to all the referee’s questions and made the necessary changes to the manuscript.

Author Response

Reviewer #2:

Comment 1: The authors have satisfactorily responded to all the referee’s questions and made the necessary changes to the manuscript.

Response : Thanks for your kind help and suggestion !
